# Prediction of Glomerular Filtration Rate Following Partial Nephrectomy for Localized Renal Cell Carcinoma with Different Machine Learning Techniques

**DOI:** 10.3390/cancers17101647

**Published:** 2025-05-13

**Authors:** Aleksander Ślusarczyk, Sumit Sharma, Karolina Garbas, Hanna Piekarczyk, Piotr Zapała, Jinhao Shi, Piotr Radziszewski, Le Qu, Łukasz Zapała

**Affiliations:** 1Department of General, Oncological and Functional Urology, Medical University of Warsaw, 02-005 Warsaw, Poland; 2Department of Urology, Jinling Hospital, Affiliated Hospital of Medical School, Nanjing University, Nanjing 210002, China; 3Department of Urology, Jinling Hospital, Jinling School of Clinical Medicine, Nanjing Medical University, Nanjing 210002, China

**Keywords:** partial nephrectomy, renal cell carcinoma, renal insufficiency, chronic kidney disease, nephron-sparing surgery

## Abstract

Preservation of kidney function is crucial for patients undergoing surgery for localized renal cell carcinoma (RCC). Partial nephrectomy (PN) is a preferred approach as it removes the tumor effectively and, in most patients, spares kidney function. However, prediction of long-term renal function after PN remains challenging. In this study, we developed a model to estimate renal function at one year after PN using easily available preoperative clinical factors. We found that preoperative glomerular filtration rate (GFR), tumor size on imaging, and comorbidity status predicted 1-year GFR. We tested various machine learning techniques but found that a conventional linear regression model performed just as well as more complex methods. Our findings suggest that simple preoperative factors can reliably predict postoperative kidney function, helping urologists to better counsel patients undergoing PN.

## 1. Introduction

Partial nephrectomy (PN) has become the standard surgical treatment for localized renal tumors [1]. Specifically, nephron-sparing surgery (NSS) is currently recommended by clinical guidelines for cT1 tumors and is feasible for cT2 renal masses [1]. The complexity of PN is influenced by tumor size, its invasiveness and location [2]. PN offers several advantages over radical nephrectomy (RN) due to the preservation of renal parenchyma. PN is associated with better overall survival (OS) and a lower risk of cardiovascular mortality than RN [3,4]. Importantly, PN is oncologically non-inferior to RN for cT1 renal cell carcinoma (RCC) in terms of cancer-specific survival (CSS), although only one randomized controlled trial was performed in that setting [5].

However, despite the nephron-sparing nature of PN, a subset of patients still experience progression of chronic kidney disease (CKD) stage postoperatively, but its risk is lower than following RN [4]. Factors such as warm ischemia time, the volume of resected kidney tissue, baseline renal function, and preoperative parenchymal volume are thought to influence postoperative renal function [6,7].

In this study, we aimed to develop and compare preoperative predictive models for mid-term renal function following PN for RCC using various machine learning techniques. By incorporating key clinical and surgical factors, our model provides a novel tool to enhance preoperative risk assessment and optimize patient counseling, which may ultimately guide clinical decision making and improve postoperative outcomes.

## 2. Materials and Methods

### 2.1. Data Collection

Medical records were retrieved in two tertiary academic centers to identify patients who underwent partial nephrectomy between 2010 and 2022.

Inclusion criteria included PNs for non-metastatic renal cell carcinoma and available preoperative and 1-year postoperative serum creatinine concentrations. Exclusion criteria encompassed patients who underwent hemodialysis due to end-stage CKD, who had kidney transplants, who converted from PN to RN due to intra- or postoperative complications, and those for whom preoperative tumor staging was unavailable. The selection flowchart is presented in Appendix A.

Systematic preoperative assessment included routine laboratory tests, medical history, comorbidity evaluation using validated questionnaires such as the age-adjusted Charlson Comorbidity Index (CCI) and Frailty Index, and radiological imaging.

Preoperative tumor assessment was performed using contrast-enhanced computed tomography (CT) imaging of the chest, abdomen, and pelvis. Imaging was used for clinical staging and tumor diameter calculation. Magnetic resonance imaging (MRI) of the abdomen was performed at the surgeon’s discretion for staging purposes.

Postoperative tumor staging was determined using the 8th edition of the American Joint Committee on Cancer (AJCC) system, while histological grading followed the World Health Organization (WHO) classification.

The estimated glomerular filtration rate (eGFR) was calculated using the Chronic Kidney Disease Epidemiology Collaboration 2021 (CKD-EPI) formula. Significant GFR loss was defined as a postoperative decrease of more than 30%. Upstaging of CKD was defined as an increase in the Kidney Disease: Improving Global Outcomes (KDIGO) 2012 CKD classification, which was used to classify the severity of renal insufficiency. Upstaging of CKD also included a transition from stage 3b to stage 3a.

Surgeries were performed via open or laparoscopic approaches, including robot-assisted procedures, depending on the surgeon’s discretion and experience.

### 2.2. Ethics Statement

This study was approved by the Institutional Review Board (Ethics Committee of the Medical University of Warsaw, No. AKBE/72/2021). All procedures performed during the study were in accordance with the ethical standards of the institutional research committee and complied with the 1964 Helsinki Declaration and its later amendments.

### 2.3. Statistical Analysis

Categorical variables are presented as numbers and percentages, while continuous variables are expressed as medians with interquartile ranges (IQRs). The dataset was randomly divided into training and testing cohorts in a 70:30 ratio. The median survival follow-up was calculated using the reverse Kaplan–Meier method. Univariable linear regression was used to identify significant clinical predictors of 1-year postoperative eGFR, followed by multivariable linear regression with stepwise selection of clinically relevant and statistically significant variables. Spearman correlation was checked for the variables that were statistically significant in the univariable analyses. A correlation matrix is presented in Appendix A. If a significant correlation was detected, the variable was considered for exclusion; however, not all significantly correlated variables were excluded—for instance, both the Charlson comorbidity score and GFR were retained despite their correlation due to their clinical relevance.

The final multivariable model was built using a combination of clinical judgment and statistical considerations with stepwise selection of variables. The assumptions of linear regression regarding the normal distribution of residuals and homoscedasticity were checked. We assessed the normality of linear regression model residuals using the histogram, Q-Q plot, and Shapiro–Wilk test. The multivariable linear regression model was visualized with a nomogram facilitating the calculation of 1-year postoperative GFR.

Models were constructed in the training cohort and internally validated in the testing one. The 10-fold cross-validation of the multivariable linear regression model was performed.

Other machine learning methods, including artificial neural networks (ANNs), support vector machines (SVMs), random forests (RFs), and XGBoost were evaluated for their accuracy in 1-year GFR prediction. Models were developed in the training cohort using exactly the same variables as in the linear regression analysis.

To evaluate and compare the performance of different machine learning models, we calculated several accuracy and calibration metrics, including R square (R^2^), mean squared error (MSE/Brier score), root mean squared error (RMSE), mean absolute error (MAE), calibration slope, and calibration-in-the-large (intercept). These metrics were computed on the testing dataset for each model to assess both predictive accuracy and calibration. The calibration of each model was plotted to compare estimated and actual postoperative eGFR values.

A *p*-value < 0.05 was considered as statistically significant. All statistical analyses were conducted using SAS software (version 9.4; Cary, NC, USA) and R programming language (version 4.2.1; Vienna, Austria).

## 3. Results

### 3.1. Baseline Perioperative Characteristics

Among 615 patients with RCC treated with PN, 415 had complete follow-up eGFR data and were included in the analysis (Appendix A). Overall, 418 patients were from the Polish center and 197 from the Chinese center. The median age of patients was 63 years (IQR 54–69). Overall, 141 (33.98%) patients were female, and 274 (66.02%) were male. Based on preoperative CT/MRI staging, 348 patients (83.86%) were classified as stage T1a, while 62 patients (14.94%) were in stage T1b. Only three patients (0.72%) were classified as stage T2a, and two patients (0.48%) as stage T2b. Tumor grades were distributed as follows: grade 1 (34.9%), grade 2 (49.6%), grade 3 (13.5%), and grade 4 (1.9%). The median tumor size on imaging was 3 cm (IQR 2–3.8). The median RENAL nephrometry score was 6 (IQR 5–7), and the median CCI was 4 (IQR 3–5). Detailed characteristics are presented in the Table 1.

Laparoscopic PN was performed in 200 patients (48.19%), including 40 (9.64%) robot-assisted procedures, while 215 patients (51.81%) underwent open surgery. Warm ischemia time shorter than 20 min was observed in 286 patients (68.92%), while 129 patients (31.08%) had a longer ischemia time. Postoperative complications of Clavien–Dindo grade ≥3 occurred in 31 patients (7.47%), while the remaining 384 patients (92.53%) had lower grade or no complications.

### 3.2. Renal Function

Preoperatively, patients were stratified by their GFR into CKD stages according to KDIGO 2012, resulting in 197 patients (47.47%) in stage 1, 159 patients (38.31%) in stage 2, 45 patients (10.84%) in stage 3, 13 patients (3.13%) in stage 4, and 1 patient (0.24%) in stage 5.

The median value of preoperative serum creatinine was 0.90 mg/dL (IQR 0.76–1.30), and the median estimated GFR was 88 mL/min/1.73 m^2^ (IQR 71–101). Among the cohort, 94 patients (23.68%) had preexisting chronic kidney disease, including 59 (14.2%) individuals who had preoperative eGFR < 60 mL/min/1.73 m^2^.

At discharge, median creatinine was 0.96 mg/dl (IQR 0.80–1.18), and GFR was 81.5 mL/min/1.73 m^2^ (IQR 62–97). Only 8.7% of patients experienced significant GFR loss (>30% decrease) at 1 year, and 85 (20.5%) individuals had 1-year eGFR < 60 mL/min/1.73 m^2^. At 1-yr follow-up, serum creatinine was 0.96 mg/dl (IQR 0.80–1.18), and the estimated GFR was 81 mL/min/1.73 m^2^ (IQR 66–97).

### 3.3. Pathological Outcomes

Postoperative pathological staging showed that 355 patients (85.54%) were classified as stage T1a, 50 patients (12.05%) as T1b, 6 patients (1.45%) as T2a, and 4 patients (0.96%) as T3a. The predominant histological subtype was clear cell RCC, which was observed in 323 patients (77.83%); papillary RCC was identified in 64 (15.42%) and chromophobe RCC in 28 (6.75%) patients. Negative surgical margins (R0) were achieved in 367 patients (88.43%), while 48 patients (11.57%) had microscopically positive surgical margins (R1). The median survival follow-up was 119 months (Interquartile range 73–154). Detailed functional and survival outcomes are presented in Table 2.

### 3.4. Univariable Linear Regression Analyses

Visual inspection of the residual Q-Q plot and histogram indicated that the residuals are approximately normally distributed, with minor deviations only at the tails (Appendix A), with the Shapiro–Wilk test suggesting deviation from a normal distribution (*p* < 0.01). The values of residuals versus fitted values for homoscedasticity assessment are shown in Appendix A. In the univariable linear regression, we found statistically significant associations between 1-year GFR and the following preoperative variables: age, diabetes, myocardial infarction, coronary artery disease, CCI, frailty, diameter on preoperative imaging, preoperative CKD, preoperative creatinine, and eGFR (Table 3).

Among postoperative variables, laparoscopic approach (Estimate: 11.2, *p* < 0.001), pathological tumor diameter (Estimate: −2.0, *p* < 0.05), and postoperative eGFR at discharge (Estimate: 0.74, *p* < 0.001) were found to be significantly associated with 1-year eGFR.

### 3.5. Multivariable Linear Regression with Preoperative Variables

Multivariable linear regression identified baseline eGFR (Estimate: 0.76, *p* < 0.001), tumor diameter on imaging (Estimate: −1.65, *p* < 0.01), and Charlson Comorbidity Index (Estimate: −1.95, *p* < 0.001) as independent predictors of 1-year eGFR. The model demonstrated good performance, with R-squared values of 0.68 in the training cohort and 0.67 in the test cohort. Our multivariable model was visualized with the nomogram (Figure 1). A 10-fold cross-validation of the multivariable linear regression model yielded an R-squared of 0.67 for predicting 1-year eGFR. The calibration of the model was acceptable and is presented in Figure 2A. Calibration plots showed reasonable agreement between predicted and actual outcomes, with a calibration slope ranging from 0.9 to 1.08 and an intercept between −7.6 and +6.0, indicating that the model’s predictions were well-calibrated.

### 3.6. Various Machine Learning Techniques

In the testing cohort, machine learning models, including ANN, SVM, RF, and XGBoost, did not surpass the performance of the linear regression model in predicting 1-year eGFR. In the testing cohort, the R-squared values for ANN, SVM, RF, and XGBoost were 0.68, 0.66, 0.64, and 0.55, respectively. Calibration plots for these models are presented in Figure 2B–E, illustrating their agreement between predicted and actual values. Model performance metrics (R^2^, MSE, RMSE, MAE, calibration slope, intercept) were compared and summarized in Table 4, highlighting differences in accuracy and calibration across the machine learning models.

## 4. Discussion

In our retrospective multicenter study, we aimed to develop a tool to preoperatively estimate the glomerular filtration rate at 1 year following partial nephrectomy for renal cell carcinoma. Our study yielded several key findings. First, we found that preoperative factors such as baseline eGFR, age-adjusted Charlson Comorbidity Index, and tumor diameter assessed on imaging are viable and easy-to-use predictors of mid-term postoperative renal function. Second, our nomogram, which illustrates the linear regression model incorporating these factors, can serve as a tool with acceptable accuracy and calibration, assisting urologists in preoperative risk assessment. Third, advanced machine learning methods (e.g., ANN) do not outperform conventional linear regression, suggesting that simple nomograms do not have to be superseded by more complex machine learning models. Fourth, we found that GFR preservation is successful in the vast majority of patients undergoing PN, highlighting the advantages of this therapeutic approach. Only 8.7% of patients experienced a >30% decline in GFR at 1 year after PN.

A recent systematic review reported 15 relevant studies that developed predictive models for eGFR following PN or RN for non-metastatic renal masses [6]. Multiple patient-, tumor-, and surgery-related factors were mentioned, and different statistical methods were used with different endpoints [6]. Some previous studies have also used linear regression models based on core predictive factors similar to ours, including preoperative renal function, comorbidities, and tumor size [8,9,10]. Our nomogram included age-adjusted CCI as a comprehensive measure of comorbidity burden in contrast to previous studies focusing on particular comorbidities, e.g., diabetes or heart diseases. For example, a study by Palacios et al. included diabetes as the only comorbidity contributing to the equation for postoperative eGFR, while also considering patients who underwent radical nephrectomy [10]. A study by Shum et al. encompassed diabetes, hypertension, heart disease, and stroke as relevant factors, in addition to other factors, including ipsilateral kidney volume [9].

Another prediction method and endpoint was used by Martini et al., who developed a nomogram to predict significant eGFR reduction (≥25% from baseline) in the time frame between 3 and 15 months after robot-assisted PN [11]. That nomogram was based on the Cox proportional hazards and included age, sex, CCI, baseline eGFR, RENAL nephrometry score, acute kidney injury (AKI) in patients with normal baseline renal function, and AKI on CKD [11]. The limitation of such a model lies in its reliance on postoperative variables and the definition of the predicted endpoint. A decline in eGFR of ≥30% has been linked to a significantly increased risk of CKD progression, cardiovascular events, and mortality, highlighting the clinical importance of accurately predicting postoperative renal function [12].

Intraoperative characteristics can also be significantly associated with eGFR but are less practical for preoperative counseling. Among these, the surgical approach has been frequently reported, with laparoscopy being favored in some studies, including one RCT, which aligns with our findings [13]. One RCT also demonstrated better 3- to 12-month eGFR outcomes for the laparoscopic approach compared to open surgery in patients undergoing PN for renal masses < 7 cm [13]. However, caution is needed when interpreting results from most retrospective studies on surgical modality due to potential confounding, as larger and more complex tumors are more likely to be treated with open surgery. Although predicting 1-year GFR based on intra- or postoperative factors was not our primary aim, our univariable analysis showed an association between the laparoscopic approach and better 1-year GFR, which may be influenced by selection bias. It is also highly important to contextualize the functional outcomes of PN within the wider adoption of robot-assisted procedures. However, one meta-analysis found no significant difference in postoperative eGFR between robotic and open approaches [14]. Nevertheless, there has been a rise in the use of laparoscopic and robotic techniques for renal surgeries in recent years. A nomogram, based on multi-institutional retrospective data, predicting 3-year risk of CKD upstaging following robot-assisted PN, included baseline eGFR, solitary kidney status, presence of multiple lesions, R.E.N.A.L. nephrometry score, clamping technique, and postoperative AKI [15]. In elderly patients, additional risk factors such as hypertension and non-achievement of trifecta have also been identified as independent predictors of progression to severe CKD, reinforcing the complexity of managing renal outcomes in high-risk populations [16].

A systematic review by Volpe et al. suggested a threshold of <25 min of warm ischemia time for optimal renal function preservation, highlighting that techniques minimizing warm ischemia contribute to better outcomes [17]. Off-clamp robotic surgery and enhanced visualization of renal vasculature (e.g., with indocyanine green) constitute examples of techniques minimizing the risk of renal ischemic injury [18,19]. A recent study demonstrated that preserved parenchymal volume could serve as a surrogate for ipsilateral renal function preservation [10]. Additionally, Kazama et al. suggested that tumor-related factors, such as larger tumor size, endophytic growth, and nearness properties from the RENAL nephrometry score, were associated with better parenchymal volume preservation [20].

One large study with robotic PNs demonstrated the predictive value of several factors, most importantly comorbidities, age, sex, and preoperative creatinine [21]. Similarly, we observed in univariable analyses that diabetes, coronary heart disease, past myocardial infarction, and preoperative chronic kidney disease were risk factors for worse eGFR. However, these were not included in the final nomogram, as they did not reach statistical significance in multivariable analyses, in which another collinear factor, the CCI, was statistically significant. The CCI, encompassing all of these comorbidities, was previously reported as a valuable tool for the prediction of oncological outcomes among urinary tract cancers at different stages [22,23,24]. Our study confirmed previous observations that the age-adjusted CCI is a practical tool for assessing comorbidities that contribute to functional outcomes following PN [6,11,25].

Advanced machine learning models, including artificial intelligence, are becoming increasingly investigated for their use in predictive models leading to personalized medicine [26,27,28]. Only a few studies demonstrated the use of novel machine learning methods for the prediction of mid-term renal function following partial or radical nephrectomy [29,30,31]. A study by Oh et al. demonstrated that the gradient boost model had superior performance compared to kernel SVM, logistic regression, decision tree, k-nearest neighbor, random forest, gradient boost, AdaBoost, and XGBoost in predicting the occurrence of CKD after PN [30]. Abdallah et al. reported the feasibility of a fully automated prediction of postoperative GFR based on CT imaging and baseline GFR [31]. However, we did not observe the advantage of advanced machine learning techniques (SVM, ANN, RF, XGBoost) over conventional linear regression in the prediction of eGFR. This can be explained by the fact that the relationships between the selected clinical and imaging variables and postoperative renal function may remain largely linear and well captured by traditional models. Artificial intelligence models may demonstrate superior performance in tasks involving image analysis and highly complex or nonlinear datasets.

Several criteria must be met to ensure the clinical application of the model. Our proposed model includes viable, cheap, and easily available clinical variables that are evaluated in each patient qualified for renal surgery, performed due to suspected malignancy. The accuracy of the model is acceptable and comparable to other literature-reported linear regression-based models [8,9,10]. As it was only internally validated, we advocate for the external validation of our tool before its use.

While PN preserves renal function, it may carry a non-negligible risk of positive surgical margins (PSM), which was 11% in our cohort. PSM increases the risk of relapse, especially in patients with adverse pathological features [32]. In contrast, radical nephrectomy mitigates this oncologic risk but results in complete dependence on the remaining kidney, which may be vulnerable to future disease or dysfunction. We observed a recurrence rate of 19.7% over a long median follow-up of 119 months, which may partly explain the higher rates, as extended follow-up allows for the detection of late recurrences.

Despite its strengths, our study has several limitations. While our model applies to the general population of patients eligible for PN, it may significantly overestimate GFR in patients with preoperative renal impairment or those with a single kidney. Another limitation is the lack of adjustment for the function of the contralateral healthy kidney, which is clinically significant. However, functional assessment of the contralateral kidney is not standard practice and, if performed, can lead to additional costs and waiting lists for radiological imaging. Incorporating additional measures of renal function might improve the model’s accuracy but would reduce its clinical applicability. Furthermore, the use of radiomics to analyze imaging features would be of potential value and require further investigation [33]. Other limitations are inherent to the retrospective nature of our study. Although we included patients from two high-volume centers, the sample size remained limited. Another limitation is the lack of external validation, which needs to be performed before clinical implementation of the model.

## 5. Conclusions

Preoperative factors, including baseline eGFR, tumor size on imaging, and age-adjusted Charlson Comorbidity Index, are effective predictors of 1-year eGFR following PN in patients with RCC. Our study demonstrates that a conventional linear regression model based on preoperative variables provides acceptable accuracy for predicting renal function after PN and is not inferior to more complex machine learning techniques.

## Figures and Tables

**Figure 1 cancers-17-01647-f001:**
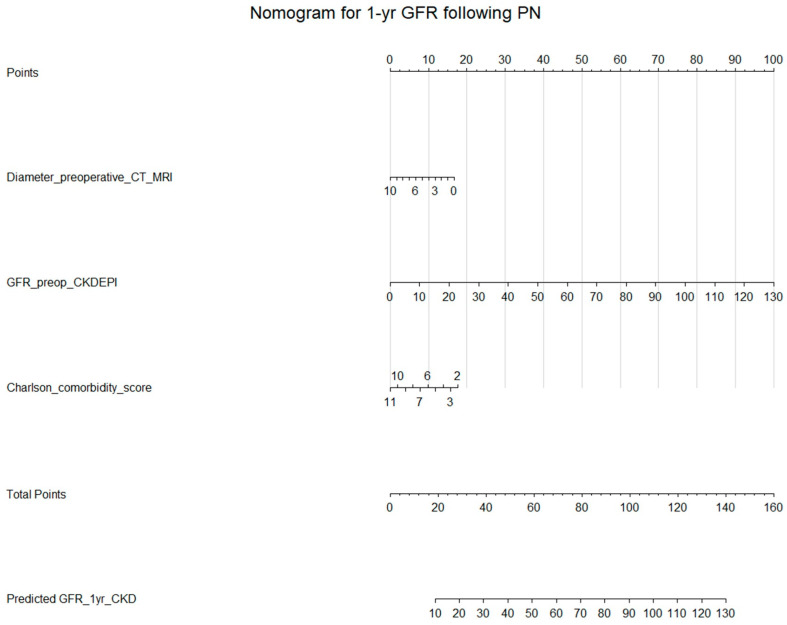
Nomogram predicting glomerular filtration rate at 1 year following partial nephrectomy.

**Figure 2 cancers-17-01647-f002:**
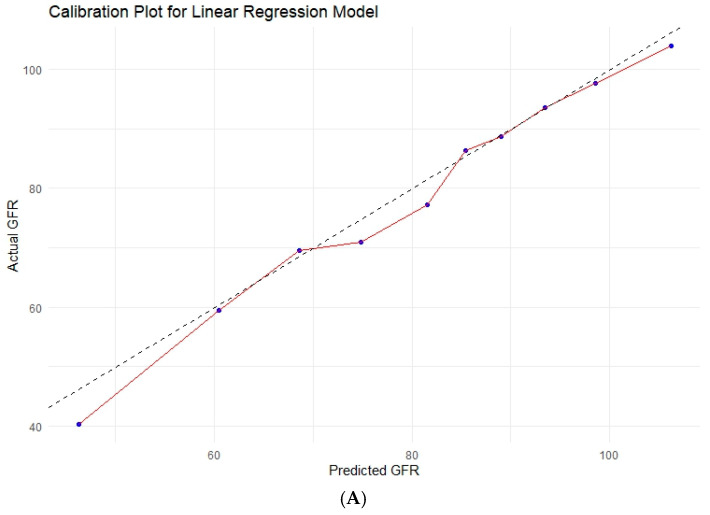
Calibration plots for the prediction of glomerular filtration rate (GFR; mL/min) at 1 year following partial nephrectomy using linear regression (**A**), artificial neural network (**B**), random forest (**C**), support vector machine (**D**), Xgboost model (**E**). The dashed line represents the ideal calibration, while the solid line represents the actual calibration.

**Table 1 cancers-17-01647-t001:** Clinical and pathological characteristics of patients treated with nephron-sparing surgery for renal cell cancer.

General Characteristics		No. of Pts./Median	%/IQR
Gender	Female	141	33.98
	Male	274	60.02
Age		63	54–69
Body mass index	kg/m^2^	25.2	23.0–28.7
Preoperative characteristics			
Clinical T stage	cT1a	348	83.86
	cT1b	62	14.94
	cT2a	3	0.72
	cT2b	2	0.48
	cT3	0	0
Tumor diameter on CT/MRI	cm	3.07	2.0–3.8
Charlson Comorbidity Index	median	4.0	3.0–5.0
Diabetes	no	345	83.13
	yes	54	13.01
	unknown	16	3.86
Hypertension	no	183	44.10
	yes	231	55.66
	unknown	1	0.24
Coronary Artery Disease	no	344	82.89
	yes	55	13.25
	unknown	16	3.86
Myocardial infarction	no	377	90.84
	yes	22	5.30
	unknown	16	3.86
Stroke	no	350	84.34
	yes	16	3.86
	unknown	49	11.80
Chronic kidney disease	no	303	73.01
	yes	94	22.65
	unknown	18	4.34
Frailty index	<18.2	284	71.36
	≥18.2	114	28.64
Postoperative characteristics			
pT stage	T1a	355	85.54
	T1b	50	12.05
	T2a	6	1.45
	T2b	0	0
	T3a	4	0.96
pN stage	N0	370	89.16
	N1	8	1.93
	Nx	37	8.92
Grade	1	145	34.94
	2	206	49.64
	3	56	13.49
	4	8	1.93
Surgical margin	negative	367	88.43
	positive	48	11.57
Tumor diameter	cm	3.0	2.0–3.9
Histology type	clear cell	323	77.83
	papillary	64	15.42
	chromophobe	28	6.75
Surgery and hospitalization			
Surgical treatment modality	Open	215	51.81
	Laparoscopic	160	38.55
	Robotic	40	9.64
Warm ischemia time	≥20 min	129	31.08
	<20 min	286	68.92
Length of stay	median	7	5–9
Complications Clavien–Dindo scale	0	194	48.74
	1	164	39.52
	2	26	6.27
	3	10	2.41
	4	4	0.96
	unknown	17	4.10

**Table 2 cancers-17-01647-t002:** Functional and oncological outcomes of nephron-sparing surgery for renal cell cancer.

Renal Function		No. of Pts./Median	%/IQR
Preoperative GFR CKD-EPI	mL/min	88.33	70.98–100.80
Discharge GFR CKD-EPI	mL/min	81.47	62.47–96.89
1-year GFR CKD-EPI	mL/min	81.0	66.0–96.89
Significant GFR loss at discharge	no	371	89.40
	yes	44	10.60
Significant * GFR loss at 1 yr	no	379	91.33
	yes	36	8.67
CKD upstage ** at 1 yr	no	292	70.36
	yes	123	29.64
Oncological outcomes			
Recurrence	no	331	79.76
	yes	82	19.76
	unknown	2	0.48
All-cause death	no	364	87.71
	yes	49	11.81
	unknown	2	0.48
Cancer-specific death	no	385	92.77
	yes	28	6.75
	unknown	2	0.48

* >30% reduction of eGFR; ** upstage of chronic kidney disease according to KDIGO 2012.

**Table 3 cancers-17-01647-t003:** Univariable (A) and multivariable linear regression (B) demonstrating the association between preoperative variables and 1-year postoperative estimated glomerular filtration rate.

A. Univariable Analysis				
Variable	Class	Estimate	Std. Error	*p*-value
Age		−0.93	0.10	<0.001
Gender	male	ref		
	female	1.47	2.87	0.608
Diabetes	no	ref		
	yes	−9.93	4.07	0.015
Myocardial Infarction	no	ref		
	yes	−12.31	5.70	0.032
Coronary Artery Disease	no	ref		
	yes	−9.60	4.08	0.019
Surgical Treatment Modality	open	ref		
	laparoscopy	11.19	2.68	<0.001
Frailty Index	<18.2	ref		
	≥18.2	−10.59	2.07	<0.001
Preoperative Tumor Diameter On CT/MRI	cm	−2.49	1.00	0.014
Preoperative Chronic Kidney Disease	no	ref		
	yes	−24.26	2.96	<0.001
Charlson Comorbidity Index	continuous	−7.29	0.68	<0.001
Preoperative GFR CKD-EPI	mL/min	0.85	0.04	<0.001
Preoperative Creatinine	mg/dL	−26.07	1.92	<0.001
Preoperative Hemoglobin	g/dL	0.12	0.08	0.147
B. Multivariable analysis				
Variable	Class	Estimate	Std. Error	*p*-value
Intercept		29.37	5.69	<0.001
Preoperative Tumor Diameter on CT/MRI	cm	−1.65	0.58	0.005
Preoperative GFR CKD-EPI	mL/min	0.76	0.04	<0.001
Charlson Comorbidity Index	continuous	−1.95	0.54	0.0004

**Table 4 cancers-17-01647-t004:** Comparison of accuracy and calibration metrics for different machine learning models predicting 1-year glomerular filtration rate following partial nephrectomy in the testing cohort.

Model	R^2^	RMSE	MAE	Brier Score (MSE)	Calibration Slope	Calibration-in-the-Large
Linear Regression	0.668	13.13	10.07	172.44	1.038	−4.700
ANN	0.679	12.92	10.15	166.88	1.076	−7.588
XGBoost	0.554	15.23	11.76	231.85	0.905	5.974
Random Forest	0.643	13.63	10.58	185.69	1.025	−3.736
SNV	0.663	13.23	10.32	175.15	1.049	−6.587

ANN: Artificial neural network, MSE: Mean Squared Error, RMSE: Root Mean Squared Error, MAE: Mean Absolute Error, Brier Score (MSE): Mean Squared Brier Score, SNV: support vector machine.

## Data Availability

The data may be obtained from the corresponding author following a reasonable request.

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
