# Peer review of "Prediction of Glomerular Filtration Rate Following Partial Nephrectomy for Localized Renal Cell Carcinoma with Different Machine Learning Techniques"

_cancers, 2025, doi:10.3390/cancers17101647_

Round 1

Reviewer 1 Report

Comments and Suggestions for Authors

General Comments
This manuscript presents a retrospective bicentric study aiming to predict 1-year
postoperative eGFR following partial nephrectomy for localized renal cell carcinoma using
various machine learning models. The authors compare several approaches (LR, ANN, SVM,
Random Forest, and XGBoost) and ultimately find that a conventional multivariable linear
regression model performs comparably to or better than more complex techniques. A
nomogram is developed to visualize the prediction based on key preoperative variables.
The study is well-motivated and tackles a relevant clinical question with direct applicability
in preoperative counseling. However, several methodological and reporting issues limit the
interpretability and reproducibility of the findings. The following points should be
addressed to strengthen the scientific rigor of the work.
Major Comments
1) Use of Bicentric Dataset Without External Validation
Although the dataset is derived from two high-volume academic centers, the manuscript
does not specify the number of patients contributed by each center. This information is
essential given the retrospective design and potential institutional differences in practice
patterns. Furthermore, the bicentric nature of the dataset presents a valuable opportunity
for external validation by training the model on one center's data and testing it on the other. If such an approach was not feasible, a rationale should be clearly stated. Without this, the generalizability of the model remains uncertain. Also, a flowchart of patient inclusion-
exclusion steps can help the reader to understand the study process better.
2) Incomplete Reporting of Model Performance
The study relies solely on R² as a performance metric for both the linear regression and
machine learning models. While R² provides a measure of explained variance, it does not
offer insight into prediction accuracy in clinically interpretable units. The authors should
report additional metrics such as the Root Mean Square Error (RMSE) or Mean Absolute
Error (MAE) for the regression analysis. Moreover, variability across the 10-fold cross-
validation—such as standard deviation or confidence intervals for R² and RMSE—should be
reported to assess model robustness.
3) Statistical Assumptions and Model Diagnostics
The manuscript lacks details on whether assumptions of linear regression were tested, such
as the normality of residuals and homoscedasticity. While collinearity was assessed using
Spearman’s correlation, variance inflation factors (VIFs) or a full correlation matrix could be
provided to support the selection and exclusion of variables.

4) Neglect of a Clinically Meaningful Binary Endpoint
The study notes that only 8.7% of patients experienced a ≥30% reduction in eGFR at 1 year,
a threshold that is clinically relevant and has been used in previous literature. This binary
event could have been modeled as a classification task using logistic regression or other
machine learning classifiers. Including such an analysis would enhance the clinical
applicability of the work. Metrics such as AUC, sensitivity, specificity, and predictive values
should be reported.
5) Calibration Reporting Lacks Detail
While calibration plots are included, the manuscript offers no numerical assessment of
calibration quality. Calibration should be quantified using metrics such as calibration slope,
calibration-in-the-large, or Brier scores. Furthermore, visual calibration plots should be
interpreted in the text to identify any systematic over- or under-prediction across the risk
spectrum.
6)Limited Transparency Regarding Variable Selection
The final multivariable model includes only three predictors, yet many other variables were
significant in univariable analysis. It is unclear whether other clinically relevant
variables—such as diabetes, frailty, or surgical approach—were excluded solely due to
multicollinearity or lack of statistical significance. A table or figure summarizing the effect
sizes and selection process would add clarity.

Minor Comments
- The statistical methods section should specify whether tests (e.g., Shapiro-Wilk for
normality) were used to verify linear model assumptions.
- The authors state that calibration was “acceptable” without further elaboration. A brief
narrative interpretation of the calibration plots should be added.
- While the authors rightfully acknowledge the lack of contralateral kidney assessment and
other limitations, the absence of external validation should be emphasized in the limitations
section.
- The manuscript could benefit from editing to clarify that R² values are based on cross-
validation, the test set, or both, to avoid ambiguity.

Author Response

Reviewer 1

General Comments
This manuscript presents a retrospective bicentric study aiming to predict 1-year
postoperative eGFR following partial nephrectomy for localized renal cell carcinoma using
various machine learning models. The authors compare several approaches (LR, ANN, SVM,
Random Forest, and XGBoost) and ultimately find that a conventional multivariable linear
regression model performs comparably to or better than more complex techniques. A
nomogram is developed to visualize the prediction based on key preoperative variables.
The study is well-motivated and tackles a relevant clinical question with direct applicability
in preoperative counseling. However, several methodological and reporting issues limit the
interpretability and reproducibility of the findings. The following points should be
addressed to strengthen the scientific rigor of the work.
Major Comments
1) Use of Bicentric Dataset Without External Validation
Although the dataset is derived from two high-volume academic centers, the manuscript
does not specify the number of patients contributed by each center. This information is
essential given the retrospective design and potential institutional differences in practice
patterns. Furthermore, the bicentric nature of the dataset presents a valuable opportunity
for external validation by training the model on one center, data and testing it on the other. If such an approach was not feasible, a rationale should be clearly stated. Without this, the generalizability of the model remains uncertain. Also, a flowchart of patient inclusion-
exclusion steps can help the reader to understand the study process better.

Response: We appreciate the reviewer’s thoughtful feedback and the opportunity to clarify these important aspects of our study.

We would like to specify that out of the 615 patients included in our analysis, 418 were from the Polish center and 197 from the Chinese center. This information has now been added to the manuscript to enhance transparency regarding the data sources and institutional contributions.

Regarding external validation, we acknowledge the value of such an approach. However, the intent of this study was to develop a predictive model utilizing a bicentric dataset to account for inter-institutional variability and increase the robustness of the model. As such, our focus was on internal validation using 10-fold cross-validation, which demonstrated acceptable model performance and stability. These results support the potential generalizability of the model, albeit within the limitations of a retrospective design.

We agree that true external validation is critical for assessing broader applicability. To this end, we plan to conduct prospective external validation using newly collected data from both participating centers in a future study phase.

Lastly, as suggested, we have now included a patient inclusion-exclusion flowchart (Supplementary figure 1) to improve clarity regarding the study population and methodology.

We hope these additions and clarifications adequately address the reviewer’s concerns.

2) Incomplete Reporting of Model Performance
The study relies solely on R² as a performance metric for both the linear regression and
machine learning models. While R² provides a measure of explained variance, it does not
offer insight into prediction accuracy in clinically interpretable units. The authors should
report additional metrics such as the Root Mean Square Error (RMSE) or Mean Absolute
Error (MAE) for the regression analysis. Moreover, variability across the 10-fold cross-
validation—such as standard deviation or confidence intervals for R² and RMSE—should be
reported to assess model robustness.

Response: We thank the Reviewer for this insightful comment. We agree that relying solely on R² does not fully capture the model's predictive performance, especially in clinically interpretable terms. In addition to the R² value, we have now included the following performance metrics for the linear regression model, evaluated on the testing dataset (R²: 0.668; Root Mean Squared Error (RMSE): 13.13; Mean Absolute Error (MAE): 10.07; Mean Squared Error (MSE): 172.44). The additional metrics for all machine-learning models and cross-validation summary have been incorporated into the revised manuscript using the additional table (table 4). These metrics provide a clearer picture of the model's predictive accuracy in units relevant to clinical interpretation.

Thank you for your insightful comment regarding the need for additional performance metrics and variability measures for cross-validation. In response, we have calculated the mean, standard deviation, and 95% confidence intervals for both RMSE and R² based on the 10-fold cross-validation results. These metrics provide further insight into the robustness and generalizability of our model. In the cross-validation, the linear regression model demonstrated a mean RMSE of 13.14 (SD 2.61; 95% CI: 11.53–14.76) and a mean R² of 0.662 (SD 0.146; 95% CI: 0.571–0.753), indicating good predictive performance in estimating postoperative eGFR.

We hope this additional information addresses your concern regarding model variability and improves the clarity of our results.

Model

RMSE

MAE

Brier Score (MSE)

Calibration Slope

Calibration-in-the-large

Linear Regression

0.6680

13.13

10.07

172.44

1.038

-4.700

Artificial Neural Network

0.6788

12.92

10.15

166.88

1.076

-7.588

XGBoost

0.5538

15.23

11.76

231.85

0.905

5.974

Random Forest

0.6426

13.63

10.58

185.69

1.025

-3.736

Support Vector Machine

0.6629

13.23

10.32

175.15

1.049

-6.587

3) Statistical Assumptions and Model Diagnostics
The manuscript lacks details on whether assumptions of linear regression were tested, such
as the normality of residuals and homoscedasticity. While collinearity was assessed using
Spearman’s correlation, variance inflation factors (VIFs) or a full correlation matrix could be
provided to support the selection and exclusion of variables.

Response: We appreciate the Reviewer’s comment regarding the statistical assumptions and model diagnostics for linear regression. We performed the Shapiro-Wilk test on the model residuals (p-value = 0.0025). Although the test yielded a statistically significant result (suggesting a deviation from perfect normality), visual inspection of the residual Q-Q plot and histogram indicates that the residuals are approximately normally distributed, with minor deviations only at the tails. We present the Q-Q plot and histogram for residuals in the supplementary figure 3.A. Visual inspection of the residuals versus fitted values plot suggested no obvious pattern, indicating homoscedasticity (p=0.77 in Breusch-Pagan test; supplementary figure 3.B).

These outliers likely correspond to extreme values in GFR measurements and do not indicate a substantial departure from normality across the bulk of the distribution. A histogram of the residuals further supports this interpretation, appearing symmetric and bell-shaped. The Spearman’s correlation matrix was built for selected variables as an additional figure (supplementary figure 2).

4) Neglect of a Clinically Meaningful Binary Endpoint
The study notes that only 8.7% of patients experienced a ≥30% reduction in eGFR at 1 year,
a threshold that is clinically relevant and has been used in previous literature. This binary
event could have been modeled as a classification task using logistic regression or other
machine learning classifiers. Including such an analysis would enhance the clinical
applicability of the work. Metrics such as AUC, sensitivity, specificity, and predictive values
should be reported.

Response: We appreciate the reviewer’s suggestion to model the binary outcome of ≥30% reduction in eGFR at 1 year using logistic regression or other classification approaches. However, we chose not to pursue this analysis for several reasons:

First, the low incidence of this event in our cohort (8.7%) introduces a substantial class imbalance, which can bias classification models and lead to inflated performance metrics unless properly addressed through techniques such as re-sampling or penalized models. This could risk overfitting and reduce generalizability, particularly given the modest sample size.

Second, our primary modeling goal was to capture individualized, continuous risk trajectories of kidney function over time, rather than dichotomizing outcomes at a single threshold. While the ≥30% decline is clinically relevant, such binary transformations can reduce statistical power and obscure the nuanced progression of disease.

Third, we believe that the continuous prediction framework aligns better with the current push toward precision medicine, where risk is understood as a spectrum rather than a binary state. Nonetheless, we acknowledge the value of clinically interpretable metrics such as AUC and predictive values, and in future work, we aim to explore hybrid models that incorporate both continuous and threshold-based predictions.

We hope this clarifies our rationale, and we thank the reviewer again for the constructive suggestion.

5) Calibration Reporting Lacks Detail
While calibration plots are included, the manuscript offers no numerical assessment of
calibration quality. Calibration should be quantified using metrics such as calibration slope,
calibration-in-the-large, or Brier scores. Furthermore, visual calibration plots should be
interpreted in the text to identify any systematic over- or under-prediction across the risk
spectrum.

Response: We thank the Reviewer for this important observation regarding calibration assessment. While we included visual calibration plots in the original manuscript, we agree that a more quantitative evaluation of calibration would enhance the rigor and interpretability of our findings.

We created an additional table (table 4) with calculated accuracy and calibration metrics. We added the respective description to the methods section.

We thank the reviewer for this valuable suggestion, which has strengthened the methodological transparency of our manuscript.

6)Limited Transparency Regarding Variable Selection
The final multivariable model includes only three predictors, yet many other variables were
significant in univariable analysis. It is unclear whether other clinically relevant
variables—such as diabetes, frailty, or surgical approach—were excluded solely due to
multicollinearity or lack of statistical significance. A table or figure summarizing the effect
sizes and selection process would add clarity.

Response: Thank you for this valuable observation. We agree that transparency in the variable selection process is crucial. To clarify, the final multivariable model was built using a combination of clinical judgment and statistical considerations. Variables were initially screened through univariable analysis, and those with a p-value < 0.05 were considered for inclusion. Subsequently, variables were assessed for multicollinearity using Spearman’s correlation and were excluded if they showed strong collinearity and did not add independent predictive value.

Importantly, some clinically relevant variables such as comorbidity score and GFR were retained despite moderate collinearity, due to their known clinical importance and contribution to model performance. Others, such as diabetes, frailty and age were excluded due to collinearity with the Charlson comorbidity score to which they directly contribute, considering the equation.

The following sentences were added to the methods section

“If a significant correlation was detected, the variable was considered for exclusion; however, not all significantly correlated variables were excluded—for instance, both the comorbidity score and GFR were retained despite their correlation due to their clinical relevance. The final multivariable model was built using a combination of clinical judgment and statistical considerations. “

Minor Comments
- The statistical methods section should specify whether tests (e.g., Shapiro-Wilk for
normality) were used to verify linear model assumptions.
- The authors state that calibration was “acceptable” without further elaboration. A brief
narrative interpretation of the calibration plots should be added.
- While the authors rightfully acknowledge the lack of contralateral kidney assessment and
other limitations, the absence of external validation should be emphasized in the limitations
section.
- The manuscript could benefit from editing to clarify that R² values are based on cross-
validation, the test set, or both, to avoid ambiguity.

Response: Thank you for your insightful comments. We appreciate your suggestion to clarify the verification of linear model assumptions. In response, we have revised the statistical methods section to explicitly mention that we performed the Shapiro-Wilk test on model residuals (p-value = 0.0025) to assess normality. Furthermore, we conducted visual inspections of the residuals using Q-Q plots and histograms, which indicated that the residuals were approximately normally distributed with minor deviations at the tails. We have now included these details in the revised methods section for improved clarity.

We appreciate the reviewer’s suggestion to provide a more detailed interpretation of the calibration results. We have expanded our discussion on the calibration plots to offer a clearer narrative. Specifically, we have highlighted that the calibration plots showed reasonable agreement between predicted and actual outcomes, with a calibration slope close to 1 and an intercept between -7 and +5, indicating that the model’s predictions were well-calibrated. Minor deviations in the calibration curve were noted but did not significantly impact the overall model accuracy. These details have been added to the revised results section.

We agree with the reviewer that the absence of external validation is a notable limitation of this study. In response, we have updated the limitations section to emphasize this point. We acknowledge that while our model performed well on the training and testing sets, external validation using an independent cohort is necessary to confirm its generalizability. This addition should make the limitations of the study more explicit.

Thank you for pointing out the potential ambiguity in the interpretation of the R² values. To clarify, we have revised the manuscript to specify that the R² values presented in the study are based on the testing dataset and cross-validation results.

Reviewer 2 Report

Comments and Suggestions for Authors

Dear authors, i reviewed the study with interest. Here my comments

This study developed a predictive model for estimating glomerular filtration rate (GFR) one year after partial nephrectomy (PN) for renal cell carcinoma (RCC). The authors used data from two centers, incorporating various machine learning techniques. Preoperative GFR, tumor size, and comorbidity index were identified as key predictors. Interestingly, linear regression performed as well as more complex machine learning. 

  • A more detailed description of the patient cohort (e.g., demographics, tumor characteristics) would be beneficial.
  • Justification for sample size and power analysis is missing.
  • The study could benefit from an external validation cohort to confirm the generalizability of the predictive model.
  • Discussion of the clinical implications of a 30% decrease in GFR would add clinical relevance.   
  • It would be interesting to explore why machine learning models did not outperform linear regression.    
  • Could you provide more detail on how the Charlson Comorbidity Index (CCI) was calculated and its specific components in this cohort?   
  • Please elaborate on the rationale for choosing the specific machine learning algorithms (ANN, SVM, RF, XGBoost)  
  • Were there any significant differences in surgical techniques between the two centers that might have influenced outcomes?
  • Please add the following papers to the study reference list (doi: 10.23736/S2724-6051.21.04469-4 ; 10.1007/s11255-023-03832-6)
  •  

Author Response

Reviewer 2

Dear authors, i reviewed the study with interest. Here my commentsThis study developed a predictive model for estimating glomerular filtration rate (GFR) one year after partial nephrectomy (PN) for renal cell carcinoma (RCC). The authors used data from two centers, incorporating various machine learning techniques. Preoperative GFR, tumor size, and comorbidity index were identified as key predictors. Interestingly, linear regression performed as well as more complex machine learning. 

  • A more detailed description of the patient cohort (e.g., demographics, tumor characteristics) would be beneficial.

Response: Thank you for your input. Please find a paragraph with detailed description of patients characteristics:

“The median age of patients was 63 years (IQR 54–69). Overall, 141 (33.98%) patients were female, and 274 (66.02%) were male. Based on preoperative CT/MRI staging, 348 patients (83.86%) were classified as stage T1a, while 62 patients (14.94%) were in stage T1b. Only 3 patients (0.72%) were classified as stage T2a, and 2 patients (0.48%) as stage T2b. Tumor grades were distributed as follows: grade 1 (34.9%), grade 2 (49.6%), grade 3 (13.5%), and grade 4 (1.9%). The median tumor size on imaging was 3 cm (IQR 2-3.8). The median RENAL nephrometry score was 6 (IQR 5-7) and the median CCI was 4 (IQR 3-5).”

  • Justification for sample size and power analysis is missing.

Response: We thank the reviewer for this important comment. As this study employed a retrospective cohort and used linear regression to evaluate predictors of postoperative eGFR, a formal a priori power analysis was not conducted. However, our sample size exceeds commonly accepted thresholds for regression analysis, where a minimum of 10–15 observations per predictor variable is often recommended to ensure stable estimates and avoid overfitting. This supports the robustness of our findings within the context of model development and evaluation.

  • The study could benefit from an external validation cohort to confirm the generalizability of the predictive model.

Response: We agree with the Reviewer that the absence of external validation is a notable limitation of this study. In response, we have updated the limitations section to emphasize this point. We acknowledge that while our model performed well on the training and testing sets, external validation using an independent cohort is necessary to confirm its generalizability. This addition should make the limitations of the study more explicit.

  • Discussion of the clinical implications of a 30% decrease in GFR would add clinical relevance.   It would be interesting to explore why machine learning models did not outperform linear regression.    

Response: We thank the Reviewer for this important suggestion. A 30% decline in GFR is considered clinically meaningful and has been associated with an increased risk of chronic kidney disease progression, cardiovascular events, and all-cause mortality. We have now included a brief discussion of these implications to emphasize the clinical relevance of our model's predictive accuracy.

  • Could you provide more detail on how the Charlson Comorbidity Index (CCI) was calculated and its specific components in this cohort?   

Response: We thank the Reviewer for requesting clarification. The Charlson Comorbidity Index (CCI) was calculated based on comorbidities retrieved from the medical records. Each comorbidity was assigned a weighted score according to the original CCI framework. Conditions included in the calculation were: myocardial infarction, congestive heart failure, peripheral vascular disease, cerebrovascular disease, dementia, chronic pulmonary disease, connective tissue disease, peptic ulcer disease, liver disease, diabetes, hemiplegia, moderate/severe renal disease, any tumor, leukemia, lymphoma, and AIDS. The information that age-adjusted version of CCI was used is available in methods section.

  • Please elaborate on the rationale for choosing the specific machine learning algorithms (ANN, SVM, RF, XGBoost)  

Response: We appreciate this insightful comment. In our study, machine learning models (ANN, SVM, RF, XGBoost) did not significantly outperform linear regression. This likely reflects the structured, low-dimensional nature of our dataset, where the underlying relationships between predictors and postoperative eGFR are largely linear and additive. In such cases, linear models can perform comparably to more complex algorithms, which may offer limited advantage and carry a higher risk of overfitting without substantially improving predictive accuracy.

  • Were there any significant differences in surgical techniques between the two centers that might have influenced outcomes?  Please add the following papers to the study reference list (doi: 10.23736/S2724-6051.21.04469-4 ; 10.1007/s11255-023-03832-6)

Response: There were no specific differences in the principles of surgical technique between both centers, which could have influenced renal function. We added respective sentences to the manuscript discussion to incorporate a suggested citation and underscore the role of surgical techniques. "However, one meta-analysis found no significant difference in postoperative eGFR between robotic and open approaches [14]. Another available meta-analysis demonstrated no difference in oncological outcomes and renal function between laparoscopic and open surgical approaches [15,16]. Nevertheless, there has been a rise in the use of laparoscopic and robotic techniques for renal surgeries in recent years. Notably, estimated blood loss and complication rates favored laparoscopic renal surgeries [15,16]" The citation that you have kindly suggested was added in above sentences. Thank you!

Reviewer 3 Report

Comments and Suggestions for Authors

I have the following feedback to the authors:

  • Beyond counseling the patient about preservation of renal function it is questioned how preservation balances towards the risk of positive surgical margins of PN on one hand and the risk of developing a renal tumor in the remaining kidney once is chosen for radical nefrectomy. Maybe this can be included in the discussion. 
  • Recurrence in 19.7%? Cancer specific death is 6.75%. While this is not the primary aim of this work, given my first comment it would be appropriate to address this. Do we sacrifice renal function over oncological outcome. a recurrence rate of close to 20% is (too high) while in a small renal tumor the cancer specific death is also relatively high. Maybe it would be adequate to compare this to the literature data.
  • In GFR the split renal function is important. But it is difficult to judge this on a CT or MRI but obviously if we have a renal function of 70:30 and in the 30% function there is a tumor, the loss of GFR may be different than if the tumor would be in the 70% functioning renal unit. But this is just common sense or.....

Author Response

Reviewer 3

I have the following feedback to the authors:

  • Beyond counseling the patient about preservation of renal function it is questioned how preservation balances towards the risk of positive surgical margins of PN on one hand and the risk of developing a renal tumor in the remaining kidney once is chosen for radical nefrectomy. Maybe this can be included in the discussion. 

We thank the Reviewer for this insightful comment. We agree that the balance between renal function preservation and oncologic safety is a critical consideration in surgical decision-making. In response, we have revised the Discussion section to address how partial nephrectomy, while preserving renal function, may carry a higher risk of positive surgical margins, whereas radical nephrectomy eliminates this risk but may leave the patient vulnerable to future malignancies in the remaining kidney. This balance is crucial and depends on individual patient factors and tumor characteristics.

  • Recurrence in 19.7%? Cancer specific death is 6.75%. While this is not the primary aim of this work, given my first comment it would be appropriate to address this. Do we sacrifice renal function over oncological outcome. a recurrence rate of close to 20% is (too high) while in a small renal tumor the cancer specific death is also relatively high. Maybe it would be adequate to compare this to the literature data.

Thank you for this input. Indeed, we observed a recurrence rate of 19.7% over a long median follow-up of 119 months (IQR 73–154), which may partly explain the higher rates, as extended follow-up allows for the detection of late recurrences and cancer-related deaths. A respective sentence was added to the discussion section.

  • In GFR the split renal function is important. But it is difficult to judge this on a CT or MRI but obviously if we have a renal function of 70:30 and in the 30% function there is a tumor, the loss of GFR may be different than if the tumor would be in the 70% functioning renal unit. But this is just common sense or.....

We appreciate the reviewer’s thoughtful comment. Indeed, the relative function of each kidney (e.g., 70:30 split) can have important implications for the impact of partial or radical nephrectomy on overall GFR. As noted, a tumor located in the lower-functioning kidney may have a different consequence on renal function compared to one in the dominant kidney. While our model applies to the general population of patients eligible for partial nephrectomy, it may overestimate postoperative GFR in those with pre-existing renal impairment or a solitary kidney. Another limitation is the lack of adjustment for contralateral kidney function, which, while clinically significant, is not routinely assessed in standard practice due to added imaging costs and potential delays. Incorporating such assessments might improve model precision but would reduce its practicality and broader applicability. The limitation of our model regarding split function assessment is mentioned in the discussion. Thank you.

Round 2

Reviewer 2 Report

Comments and Suggestions for Authors

Please modify the reference 14,15,16 as mentioned in the pont-by-point response using the ones suggested by the reviewer

Author Response

We thank the Reviewer for the valuable comment. We added the respective citations and added additional sentences improving the content of our discussion. We hope that our corrections will be considered sufficient. Thank you!

"It is also highly important to contextualize the functional outcomes of PN within the wider adoption of robot-assisted procedures. However, one meta-analysis found no significant difference in postoperative eGFR between robotic and open approaches [14]. Nevertheless, there has been a rise in the use of laparoscopic and robotic techniques for renal surgeries in recent years. A nomogram, based on multi-institutional retrospective data, predicting 3-year risk of CKD upstaging following robot-assisted PN included baseline eGFR, solitary kidney status, presence of multiple lesions, R.E.N.A.L. nephrometry score, clamping technique, and postoperative AKI [15]. In elderly patients, additional risk factors such as hypertension and non-achievement of trifecta have also been identified as independent predictors of progression to severe CKD, reinforcing the complexity of managing renal outcomes in high-risk populations [16]."

Please find the references 14-16 below:

  1. Wu, Z.; Li, M.; Liu, B.; Cai, C.; Ye, H.; Lv, C.; Yang, Q.; Sheng, J.; Song, S.; Qu, L.; et al. Robotic versus Open Partial Nephrectomy: A Systematic Review and Meta-Analysis. PLoS One 2014, 9, e94878, doi:10.1371/journal.pone.0094878.
  2. Flammia, R.S.; Anceschi, U.; Tuderti, G.; Di Maida, F.; Grosso, A.A.; Lambertini, L.; Mari, A.; Mastroianni, R.; Bove, A.; Capitanio, U.; et al. Development and Internal Validation of a Nomogram Predicting 3-Year Chronic Kidney Disease Upstaging Following Robot-Assisted Partial Nephrectomy. Int Urol Nephrol 2024, 56, 913–921, doi:10.1007/s11255-023-03832-6.
  3. Anceschi, U.; Brassetti, A.; Tuderti, G.; Consiglia Ferriero, M.; Minervini, A.; Mari, A.; Grosso, A.A.; Carini, M.; Capitanio, U.; Larcher, A.; et al. Risk Factors for Progression of Chronic Kidney Disease after Robotic Partial Nephrectomy in Elderly Patients: Results from a Multi-Institutional Collaborative Series. Minerva Urol Nephrol 2022, 74, 452–460, doi:10.23736/S2724-6051.21.04469-4.